# Early Melanoma Detection Based on a Hybrid YOLOv5 and ResNet Technique

**DOI:** 10.3390/diagnostics13172804

**Published:** 2023-08-30

**Authors:** Manar Elshahawy, Ahmed Elnemr, Mihai Oproescu, Adriana-Gabriela Schiopu, Ahmed Elgarayhi, Mohammed M. Elmogy, Mohammed Sallah

**Affiliations:** 1Information Technology Department, Faculty of Computers and Information, Mansoura University, Mansoura 35516, Egypt; manarelshahawy@mans.edu.eg; 2Applied Mathematical Physics Research Group, Physics Department, Faculty of Science, Mansoura University, Mansoura 35516, Egypt; ahmedelnemr@mans.edu.eg (A.E.); elgarayhi@mans.edu.eg (A.E.); 3Faculty of Electronics, Communication, and Computer Science, University of Pitesti, 110040 Pitesti, Romania; 4Department of Manufacturing and Industrial Management, Faculty of Mechanics and Technology, University of Pitesti, 110040 Pitesti, Romania; gabriela.plaiasu@upit.ro; 5Department of Physics, College of Sciences, University of Bisha, P.O. Box 344, Bisha 61922, Saudi Arabia; msallahd@mans.edu.eg

**Keywords:** skin cancer classification, melanoma detection, you only look once (YOLO), dermatoscopic image analysis, ResNet50 network

## Abstract

Skin cancer, specifically melanoma, is a serious health issue that arises from the melanocytes, the cells that produce melanin, the pigment responsible for skin color. With skin cancer on the rise, the timely identification of skin lesions is crucial for effective treatment. However, the similarity between some skin lesions can result in misclassification, which is a significant problem. It is important to note that benign skin lesions are more prevalent than malignant ones, which can lead to overly cautious algorithms and incorrect results. As a solution, researchers are developing computer-assisted diagnostic tools to detect malignant tumors early. First, a new model based on the combination of “you only look once” (YOLOv5) and “ResNet50” is proposed for melanoma detection with its degree using humans against a machine with 10,000 training images (HAM10000). Second, feature maps integrate gradient change, which allows rapid inference, boosts precision, and reduces the number of hyperparameters in the model, making it smaller. Finally, the current YOLOv5 model is changed to obtain the desired outcomes by adding new classes for dermatoscopic images of typical lesions with pigmented skin. The proposed approach improves melanoma detection with a real-time speed of 0.4 MS of non-maximum suppression (NMS) per image. The performance metrics average is 99.0%, 98.6%, 98.8%, 99.5, 98.3%, and 98.7% for the precision, recall, dice similarity coefficient (DSC), accuracy, mean average precision (MAP) from 0.0 to 0.5, and MAP from 0.5 to 0.95, respectively. Compared to current melanoma detection approaches, the provided approach is more efficient in using deep features.

## 1. Introduction

The skin is the outermost layer of the human body. The largest organ in the human integument structure comprises multiple layers. It also involves immune cells and cells that generate melanin to keep the body healthy from the carcinogenic potential of ultraviolet radiation [1]. Skin cancer arises when skin cells become disorganized and proliferate uncontrollably, potentially migrating to other body parts. Skin cancer is the most common type of cancer worldwide. Melanoma and non-melanoma pigmented lesions are the two main forms of skin cancer. Melanoma is associated with melanocytes, influencing the color of malignant cells [2].

In 2023, the American Cancer Society predicts that about 97,610 new melanomas will be diagnosed in the country [3]. Melanoma is predicted to be fatal to approximately 7990 people (about 5420 men and 2570 women). Rates of melanoma have considerably increased in recent decades. On the other hand, it is more deadly since it can spread to many other body areas if it is not detected and treated promptly. In addition, it is associated with melanocytes, which cause malignant cells to change color, thus accounting for the bulk of pigmented skin cancer deaths [4]. Figure 1 depicts various types of skin cancer lesions, including common types detected.

Due to the substantial similarities between the many forms of skin lesions, a visual analysis is challenging, which can lead to incorrect choices [5]. The ABCD (asymmetry, border irregularity, color variation, and diameter) examination is commonly used to diagnose malignant melanoma. Patients with melanoma who are detected early have a greater chance of survival [6]. As digital computing capabilities progress, some researchers have developed computer-aided diagnosis (CAD) systems that integrate image processing, pattern recognition, and artificial neural networks to support physicians in diagnosing [7].

According to Figure 2a,c,e,g, they are considered benign, but Figure 2b,d,f,h are melanoma skin cancer. The following is because of the result of the ABCD examination: 

The ABCD approach was utilized to distinguish malignant melanoma from benign lesions based on its ability to extract distinguishing morphological features. As a result, it is usually utilized in automated systems. This method’s efficiency ranges from 85.0% to 91.0%. These low percentages inspire researchers to provide another approach, either by altering an existing method or developing a new approach to improve performance.

Deep learning (DL) techniques must be the most effective, supervised, time-consuming, and cost-efficient machine learning method. DL techniques have substantially advanced in automatically extracting characteristics across several deep layers, generating significant benefits [8,9]. DL has recently been successfully employed by academics worldwide in visual tasks and object recognition.

### Characteristics of Lesions

The characteristics of melanoma have been used to construct certain machine learning approaches to identify the disease. CAD models involve effective algorithms to categorize and forecast melanoma. Algorithms like the adaptive histogram equalization approach, contrast stretching, and a median filter are used to improve the pictures. Following that, there are a variety of segmentation algorithms, including normalized Otsu’s segmentation (NOS), which separates the damaged skin lesion from the normal skin and solves the issue of fluctuating illumination [6]. The segmented images are used to construct and extract features, which are then given to the various classifiers, including hybrid Adaboost, a support vector machine (SVM), and DL neural networks [10,11,12,13,14]. Various architectures such as ResNet, Dense Net, and Senet were used. Various methods are applied to deal with each class’s unequal quantity of images, such as balanced batch sampling and loss weighting.

Melanoma is a serious form of skin cancer that can be deadly. Detecting it in its early stages is crucial for successful treatment and improved health outcomes. The primary goal of melanoma detection is to increase early detection rates, which could potentially save lives. However, it can be difficult for individuals to identify melanoma independently because it can resemble other skin lesions, such as moles.

The proposed system was pre-trained more than once with various hyperparameter settings to obtain an improved consistent skin lesion classification technique. The suggested architecture uses a single stage to combine detection and classification instead of more conventional approaches. The proposed system has four significant contributions over previous computer-assisted skin cancer screening approaches:The proposed method applies to any image (dermoscopy or photographic) of pigmented skin lesions using you only look once (YOLOv5) and ResNet.The suggested system classifies samples and determines each class with probability.It interacted directly with the skin-color images that were obtained with different sizes.The proposed approach enhances scalability since YOLO and ResNet can detect melanoma in huge datasets. This is crucial as it allows for developing more precise and effective melanoma detection systems.

This study developed, implemented, and effectively assessed a novel DL-based skin lesion classification algorithm against a publicly available dermoscopy dataset based on utilizing seven categories of skin lesions (the HAM10000 dataset) [15].

The arrangement of the paper is outlined below. The second section offers a review of related studies. Section 3 goes into detail about the suggested method and the datasets utilized. Section 4 outlines the experimental setup and performance assessment of the suggested technique, and Section 5, in the end, discusses the conclusions.

## 2. Related Work

The early detection and treatment of melanoma frequently result in a cure. It becomes more dangerous and difficult to treat if it penetrates deeper into the skin or other body parts. Most melanoma classification algorithms currently in use contain custom-made characteristics such as measurements of lesion shape, distribution, and color, as well as measurements of texture and border irregularity [16]. After feature extraction, machine learning techniques such as artificial neural networks (ANNs), K-nearest neighbors (KNN) classification, SoftMax classification, SVM, and logistic regression can be used to successfully solve the classification problem [17]. Through deep learning models, it is possible to train machines to provide personalized treatment, classify medical images (like X-rays), identify new biomarkers, and predict patient outcomes such as the risk of death or the success rate of surgery. Additionally, this technology can potentially reduce healthcare costs [18,19]. We provided examples of pertinent DL research in Table 1.

Using a contrast-constrained adaptive histogram equalization strategy, Premaladha et al. [6] improved the melanoma classification system. The images were enhanced before segmenting the filtered grayscale image using the Otsu normalized method. DL achieved a 92.8% classification accuracy. A deep convolutional neural network (DCNN) divided color images of skin cancer into three groups: atypical nevus, melanoma, and typical nevus from Med node and PH2 datasets. This proposed system needs to add more classes for more accuracy.

Codella et al. [10] established a hybrid technique for melanoma categorization. A help vector was used in this method. A support vector machine (SVM), deep learning, and sparse coding were used. In total, 2624 clinical cases from the International Skin Imaging Collaboration were used as a dataset. When all the results were added together, the categorization efficiency was 93.1%. Deepening feature extraction and adding more cases to diagnose melanoma are needed.

Gessert et al. [11] used a huge ensemble of state-of-the-art convolutional neural network (CNN) models to classify skin lesions. Various architectures such as ResNet, Dense Net, and Senet were used. They applied various methods to deal with the unequal quantity of images for each class, such as balanced batch sampling and loss weighting. Finally, the ensemble of multiple convolutional neural network architectures was fine-tuned utilizing their dataset. This task was correctly classified 85.1% of the time.

Waheed et al. [12] constructed a model using machine learning for diagnosing melanoma depending on dermoscopy images from the HAM10000 dataset. It was based on distinguishing attributes such as the appearance and texture of many skin lesions. SVM was used in their research to distinguish melanoma images from all other classes. Their model correctly classified objects 96.0% of the time. There is a need for more attributes of skin lesion classification.

To categorize skin lesion images into five diagnostic groups, Hekler et al. [20] used CNN. They applied their method to 300 test images from the HAM10000 dataset and discovered that it was 82.9% accurate (60% for each of the five illness classifications). An invasive technique was demonstrated with a small number of low-resolution pictures. Their method’s binary distinction between melanoma and nevi is another drawback.

DCNN evaluation for melanoma categorization was given by Pham et al. [21]. Additionally, they helped with data improvement. Using CNN-level layers at different levels, each feature was retrieved. Additionally, the dataset was probably altered in some way. The analysis was tested using the ISBI dataset, which had an area under the curve (AUC) of 89.2%. Because skin lesion images were identical, it was necessary to reuse network weights to increase sensitivity.

Yu et al. [22] published a two-stage melanoma detection approach. They used a deep residual network (DRN) for classifying and a DCNN network with more than 50 layers for segmentation. Segmentation was performed using a fully convolutional residual network (FCRN), while classification was performed using a DRN. The ISBI dataset evaluated the findings, and the AUC was 80.4%. It is vital to explore techniques to include Bayesian learning, particularly probabilistic graphical models, in networks to further increase the discrimination capability of the very deep CNNs and address the issue of insufficient training data.

Li and Shen [23] proposed an automated melanoma detection method based on two deep learning techniques. They employed two FCRNs simultaneously for a more thorough classification. The lesion feature network was used to extract dermoscopy features. The model used the International Skin Imaging Collaboration (ISIC 2017) dataset to test the performance. It produced 2357 photos of both malignant and harmless oncological illnesses for this collection (ISIC). All pictures were sorted based on the categories identified using ISIC, and the same number of photos were used to divide each subgroup. Model segmentation and classification results were 75.3% and 91.2%, respectively. There was overfitting in AUC, and the results of segmentation were low.

Seeja and Suresh [24] presented a DCNN for precise skin lesion segmentation using the U-net technique. To obtain their findings, they combined CNN and FCNN. The color, texture, and shape attributes were selected from the segmented images of the ISBI 2016 dataset. The method used for a texture analysis involved a local binary pattern (LBP). Form features were extracted using the edge histogram, Gabor, and histogram of oriented gradients (HOG) approaches. For classification, SVM, random forest (RF), K-nearest neighbor (KNN), and naïve Bayes (NB) classifiers were selected. The test results showed that the dice co-efficiency value for image segmentation was 77.5%. The classification accuracy of the SVM classifier was 85.1%, 82.2% for RF, 79.2% for KNN, and 65.9% for NB. The proposed system needs to improve the result for classification.

Nasiri et al. [25] created a method of case-based reasoning for early melanoma detection. DL algorithms were used to categorize skin lesions in their strategy. This investigation is a follow-up to their case-based learning assistant system study that looked at how to detect and predict melanoma from the ISIC dataset. A 19-layer model of CNN, a deep learning method, was used in this work to categorize skin lesions. Three fully connected layers, three max-pooling layers, and eleven convolutional layers form the model. The proposed approach had a 75.0% success rate in the ISIC Melanoma Project, where it was tested.

Inthiyaz et al. [26] recently created an approach based on combining CNN and SoftMax. Skin photographs were first filtered to remove unwanted noise from the image before being processed to improve the overall quality of the image. The presented work was based on extracting features from skin images, which were then classified using the SoftMax classifier. It was accurate to 87.0%. This model should be improved for accuracy by including more skin cancer classes and employing DL methods.

Huang et al. [27] utilized hyperspectral imaging to examine the ISIC dataset and applied YOLOv5 to identify and categorize various types of skin cancer. The focus of the training phase was on three categories: basal cell carcinoma (BCC), squamous cell carcinoma (SCC), and seborrheic keratosis (SK). However, the model may have developed a bias toward detecting skin cancer within a specific demographic. The accuracy rates for the RGB and HSI models were 79.2% and 78.7%, respectively. The proposed model used hyperspectral imaging to eliminate noise, but it was insufficient. Additionally, there were resemblances between BCC and SK, which resulted in confusion between the categories and lower accuracy.

Considering the importance of early melanoma identification, the visible similarity among melanoma and non-melanoma tumors, the absence of contrast between lesions and the skin, and other considerations, therefore, the accurate automatic diagnosis of skin tumors is crucial to improve the precision and effectiveness of pathologists.

According to previous studies, there are limitations in melanoma detection due to the small number of classes used and the need to determine the degree of classes. Deepening feature extraction and adding more cases to diagnose melanoma are needed. The similarities between classes cannot be determined. Recently, work has been based on two stages to determine melanoma (segmentation and classification). But there is overfitting in AUC, and the segmentation results are low. A system that relies on DL must be created to obtain reliable melanoma classification. YOLOv5 relies on a single step for identifying and classifying skin lesions to determine the type of melanoma, in contrast to earlier DL studies for skin lesion classification that concentrated on employing specific layers. The core concept behind the YOLO technique is to employ an end-to-end convolutional neural network to predict the target’s class and position. It uses bounding boxes for detection and probabilities to determine an object’s probability percentage. The design is composed of an input layer, a convolutional layer, a layer for pooling, a layer for fully connecting, and a layer for the output. In the primary step, YOLO splits the input image into S × S grids. Each grid is diagnosed to check if it has any class of skin lesions. It then classifies each object and gives it its probability. Then, the ResNet network is used to prevent gradient explosion issues. Because we have seven classes with multiple scales, it is employed as the image classification network.

## 3. Proposed Melanoma Detection Technique

### 3.1. Dataset

If the dataset is limited and the data does not contain various images, i.e., photos of different classes, training neural networks on them is extremely difficult. If the dataset is heavily skewed, it will not meet the goal and may give us an incorrect impression of accomplishment. Fortunately, the HAM10000 dataset is utilized. The HAM10000 dataset has been made available to the public to aid dermatoscopic image recognition research. 10,015 dermatoscopic images from the ISIC collection make up the HAM dataset. Multiple procedures are used to obtain HAM dermatoscopic images from varied populations. This dataset can be used to identify benign keratosis lesions (BKL), melanoma (MEL), vascular lesions (VASC), basal cell carcinoma (BCC), actinic keratosis (AKIEC), dermatofibroma (DF), and melanocytic nevi (NV). These classes are added to the data configuration file, and the number of classes is changed to seven. Most of these lesions are confirmed with histopathology. The dataset is split into two sets, as listed in Table 2.

### 3.2. Experimental Platform

The YOLOv5 algorithm is trained using the HAM10000 datasets in Google Collaboratory (Google Colab), a free integrated development environment (IDE). Jupyter notebooks are hosted for machine learning and data science researchers to contribute to reproducible experiments and technique descriptions. The key advantage is that it enables researchers with the computational power to run recent DL approaches interactively, eliminating the need to configure software packages and dependencies separately. Tesla K80 with two cores is utilizing Google Colab based on the Linux platform with mostly 12 GB for RAM, which Google gives to facilitate the machine learning (ML) training and analysis. It is appealing because Google Colab has pre-installed libraries, as addressed in Table 3. It helps the DL model become more accurate and access larger datasets.

### 3.3. The Structure of the YOLOv5-S Model

The YOLO model is a target detection method that uses regression. A regression model is created from the target detection problem. When photos of skin cancer are entered into a DNN using YOLO, the technique predicts the classification and localization information of the various skin lesion classes based on the computation of the loss function [28]. YOLOv5 is based on the YOLO detection architecture. It employs top-notch algorithm optimization techniques developed in recent years in convolutional neural networks, including auto-mosaic data augmentation, learning bounding box anchors, the Leaky Rectified Linear Unit (Leaky Relu) activation function, and others. They oversee various tasks in various parts of the YOLOv5 architecture, as shown in Figure 3.

In the architecture, YOLOv5 comprises the dataset images, backbone, neck, and detection output components, as shown in Figure 4. The input is the preprocessing output according to the previous subsection. The second component is the backbone. This backbone eliminates the redundant gradient information present in large backbones. Gradient change is incorporated into feature maps, which speeds up inference, improves accuracy, and shrinks the model’s size by reducing the number of hyperparameters. It is a CNN that uses the Cross-Stage Partial network (CSP) and focuses the interlaced sampling splicing structure as its core to produce significant features from provided pictures. The problem of recurring gradient information in large-scale networks is addressed with CSPNET [29]. Lowering model hyperparameters and FLOPS (floating-point operations per second) decreases calculations while increasing the speed and precision of inference and shrinking the model’s size.

Second, feature pyramids are created using the neck model. Models can achieve good object scaling generalization with the help of feature pyramids. It helps with object identification when it appears in different scales and sizes. The neck model of YOLOv5 is based on spatial pyramid pooling (SPP) and a path aggregation network (PANET) [30,31]. It increases the utilization of precise location signals at lower layers and information flow, increasing the accuracy of object location. The spatial scale of the convolved information is decreased in this version due to the employment of a subsampling layer. By lowering dimensionality, the amount of computing required to process the data should be minimized.

The YOLO layer completes the last detection step, which is the head of YOLOv5. The method can simulate small, medium-sized, and large objects according to the generation of a multi-scale prediction for the anchor boxes. To optimize the overlap between the ground truth and the anticipated bounding box of the detected class, a generalized intersection over union (GIOU-loss) is utilized [32]. Stochastic gradient descent (SGD) is used by default in the original version of YOLOv5 [33]. It is a straightforward yet highly effective method for fitting linear classifiers with convex loss functions. SGD rapidly produced divergence at the specified learning rate of 0.0001. After training started, loss parameters grew significantly and eventually reached infinity after around ten batches. So, Adam is used as an optimizer in the training step since it consistently converges. With ADAM serving as the adaptive learning rate, it begins with an initial learning rate.

To explain how a certain class is used, the network’s final layer employs SoftMax. The training stage receives the detection stage’s results after being altered. These findings include the bounding box coordinates $b_{x},b_{y},b_{w},b_{h}$ and c is the class of the detected class and represented in a vector as [$p_{c},b_{x},b_{y},b_{w},b_{h},c]$ as well as the probability of each class $p_{c}$ for each detection. The intersection over union (IOU) represents the accuracy of the target skin cancer class anticipated and the actual skin cancer class. When there is no object in the target image, the probability of detecting an object is 0. When there is a complete object, the probability is equal to 1, and the IOU is calculated using the predicted target (p) and the real target (t) as Equation (1).
(1)$${IOU}_{p}^{t}=\frac{{box}_{p}⋂{box}_{t}}{{box}_{p}⋃{box}_{t}}$$

The real target boundary is box_t_, whereas the expected target is box_p_. As observed, the IOU is the ratio of the intersection of the predicted and real target frames to their union. Then, each image’s annotations are recorded in YOLO format in Txt files, with each line containing a skin lesion bounding box description. The training stage is then conducted using YOLOv5. Once the Yolov5 original model has been applied, an additional layer is applied to increase efficiency. A new scale includes convolutional layers with activation functions to enhance object detection.

Target classification uses ResNet as a tool. The issue of the network advancing more deeply without gradient explosion is resolved with the formation of the ResNet network. As is well known, DCNN excels at extracting low-, medium-, and high-level characteristics from images. Accuracy is improved by stacking more layers. The residual module, which comprises two dense layers and a skip connection, is the main point of ResNet. Each of the two dense layers has a different RELU function that activates it.

### 3.4. Preprocessing

The preprocessing step of melanoma detection aims to provide a suitable source for the model’s appropriateness in actual time. Data augmentation is provided to increase the sample size for those imbalanced classes and prevent this imbalanced dataset from producing a biased or skewed prediction. Since an image could contain noise, preprocessing is necessary for detection algorithms [34,35]. Pictures of skin lesions often have uneven lighting, skin surface light reflection, and hair. These kinds of noises need to be reduced because they can impair segmentation performance.

Additionally, each of these photographs was carefully scaled and smoothed. Normalizing the original image speeds up detection without sacrificing any data. This is an essential technique to guarantee that every image is annotated and to increase performance accuracy. It is performed for the necessary computation parameters as well as for later implementation and detection. The DL architecture scales image pixels before the training process. In experiments, photos are scaled to (224, 224, 3) using the ImageDataGenerator class and scaling methods. The image pixels are normalized to standardize the image samples. The [0, 255] range of possible values is replaced with the [0, 1] range. If images are not scaled, they will receive many votes to update weights due to their wide pixel range. The YOLO model provides the output from the processed stage [36,37]. After combining the detection model’s anchor box size for classification, the preparation methods would be completed according to Figure 3. There are many models of YOLOv5.

The proposed framework utilized a small model with only 7 MB of memory. Here is a brief explanation of all the versions of YOLOv5, which is used for model configuration. First, YOLOv5n (the nano edition) is the smallest in the series, intended for Internet-of-Things data, and it also supports OpenCV Dl. In the INT8 format, it weighs less than 2.5 MB; in the FP32 format, it is about 4 MB. It is perfect for software devices. Second, YOLOv5s (the small edition) is one of the family’s smallest models, with about 7.2 million parameters, making it perfect for inference to run on the CPU. Third, this medium-sized version is interesting, YOLOv5m (the medium edition), which has 21.2 million parameters. Given that it offers a pretty good balance between speed and efficiency, it may be the model that is most suitable for many datasets and training. Fourth, YOLOv5l (the large edition) contains 46.5 million components, one of the biggest versions in the YOLOv5 group. For datasets that necessitate us to find tiny things, it works perfectly. Finally, YOLOv5x (the extra-large edition) is the largest of the five and has the greatest MAP, although it has 86.7 million parameters and therefore is slower than the others. But ResNet has 25.6 million parameters. The network has a performance capability of 17 GFlops. Gigaflops is a unit of measurement used to measure the computational power of a computer’s floating-point unit.

## 4. Experimental Results

### 4.1. Performance Metrics

The evaluation outcomes of the trained model are computed using several performance indicators. Precision, often referred to as the positive predictive value, is defined in Equation (2). It is a metric used in this study to show how well the model detects skin lesions. The recall rate in Equation (3) is also called sensitivity. It is determined by dividing the number of skin lesions genuinely detected by the number of skin lesions that are discovered and missed in each image. The harmonic mean of recall and precision is used to determine the dice similarity coefficient (*DSC*) using Equation (4). Specificity is also calculated, which is the true negative rate using Equation (5), and accuracy is calculated by using Equation (6). As shown below, false negatives (*FN*), false positives (*FP*), true negatives (*TN*), and true positives (*TP*) are all used to calculate them.
(2)$$Precision=\frac{TP}{TP+FP}$$
(3)$$Recall/Sensitivity=\frac{TP}{TP+FN}$$
(4)$$DSC=\frac{2*TP}{2*TP+FP+FN}=2*\frac{Precision*Recall}{Precision+Recall}$$
(5)$$Specificity=\frac{TN}{TN+FP}$$
(6)$$Accuracy=\frac{TP+TN}{P+N}=\frac{TP+TN}{TP+TN+FP+FN}$$

By averaging the average precision (*AP*) of each class, the mean average precision (*MAP*) is obtained using Equation (7). It is frequently used to assess how well object detection algorithms function. The MAP formula comprises various submatrices, including the confusion matrix, GIOU, recall, and precision. The detection model’s ground truth bounding boxes overlap the anticipated and actual objects, and GIOU measures this overlap. Each IOU threshold value results in a unique MAP. Therefore, this value needs to be provided. An IOU is compared to a defined threshold, generating either a correct or incorrect detection. The performance of the trained algorithm is assessed using GIOU criteria of 0.5 and 0.5:0.95 to examine the efficiency of the skin lesion model in a set of experiments.
(7)$$MAP=\frac{1}{N}\sum_{N}^{k=1}AP_{k}$$

### 4.2. Results

Images are given learnable weights and biases in this stage of preprocessing. The YOLOv5 algorithm is then utilized with the initial configuration presented in Table 4. Because of its quick execution, it is possible to use the YOLO-trained model in real-time with a prediction in a split second. In the first run, the 12,519 dermoscopy photos from seven different types of skin cancer in the HAM10000 dataset were split into training and testing sets, with training sets accounting for 80% of the dataset’s total data and testing sets accounting for 20%. The HAM dataset is tested over seven classes (BKL, AKIEC, VASC, BCC, DF, NV, and MEL) using 9514 dermoscopy images from the original training run and 3005 from the testing run. The network is trained for a total of 300 epochs. Each class’s six performance measures are generated separately to assess the proposed method’s performance. As a result, the average of these values is computed. The performance metrics average is 98.1%, 97.5%, 97.7%, 98.9%, 97.5%, 97.1%, and 96.3% for precision, recall, DSC, specificity, accuracy, MAP from 0.0 to 0.5, and MAP from 0.5 to 0.95, respectively. With the help of the settings in Table 4 for the first experiment run, Table 5 displays the metrics’ findings for the HAM10000 dataset.

The same splitting ratio of 80% training and 20% testing was used for the second run, which used new parameters. The new parameters are 100, 32, and 0.0001 for the epochs, batch size, and learning rate, respectively. The greatest results were achieved with a batch size of 32. So, the performance metrics average is 99.0%, 98.6%, 98.8%, 99.5%, 99.8%, 98.3%, and 98.7% for precision, recall, DSC, accuracy, specificity, MAP from 0.0 to 0.5, and MAP from 0.5 to 0.95, respectively, as in Table 6.

It is possible to determine the sensitivity of the neural network in Figure 5 using the MAP to summarize the study’s findings.

The precision and recall curves in Figure 5a,b are implemented at a network size and evaluated at the GIOU threshold range from 0.5 to 0.95. The model worked well throughout, with the greatest MAP from 0.5 to 0.95 value of 98.7% occurring for the network size 224 with a threshold value equal to 0.5. Additionally, the weights produced with YOLOv5 (S-Model) require 14 MB.

The MAP determines the area under the precision–recall curve, making it a useful tool for comparing various models regardless of the confidence level. Figure 5a,b illustrates how recall and precision grow with increasing epochs. Additionally, when the confidence score for each class differs, the effectiveness of the melanoma detection performed with the YOLOv5 models is evaluated by looking at the precision–recall curve. When precision retains a considerable contribution to growth in the recall, it is easier to evaluate the capacity to predict melanoma. The goal is to find the confidence level that maximizes F1 across all classes. In this case, the results are shown in Figure 6 with a confidence of 70.8%, a precision of 96.0%, and a recall of 91.0%. In the illustration in Figure 7, batch selections are made from the testing set to display the bounding box for each class with each probability.

## 5. Discussion

The early detection of melanoma is crucial for improving treatment and prognosis. Screening a vast number of images for melanoma can be achieved using YOLOv5 and ResNet, which can enhance early detection rates. Through this, more lives can be saved, and the quality of life for melanoma patients can be improved. A comparison analysis is presented in this section to assess the performance of the suggested model as a melanoma detection approach utilizing modified YOLOv5 and ResNet techniques. In Table 7, our model is compared with other models. Despite the similarities between classes, the suggested YOLOv5 detected melanoma using adequate coordinates, including the bounding box. The comparison results allow us to conclude that the suggested YOLOv5 model is reliable for melanoma detection in real-time photos that have been gathered.

The results show that our model achieves more accuracy, a better performance, and a more accurate network. In the comparison, studies used different methods with different and the same datasets. For the HAM10000 dataset, Ali et al. [39] achieved 91.9% using CNN, and Khaledyan et al. [40] achieved 83.6% using Ensemble Bayesian Networks for the precision measure. In addition to these references, Alsaade et al. [38] produced a model using CNN based on the PH2 dataset, which contains 40 melanomas, 80 normal nevi, and 80 abnormal nevi, as in Table 8. The model achieved 97.5% accuracy. Chang et al. [41] accomplished 94.1% using the XBG classifier using 10-fold cross-validation. Despite this, our model achieves its best performance using two-fold cross-validation. Kawahara et al. [42] used 1700 photos from the ISIC-ISBI 2017 skin analysis challenge, which were used to train our network, and 300 images were utilized to assess the network’s efficiency using various hyperparameters. It exhibited a fully convolutional neural network that could extract clinical dermoscopic features from photos of dermoscopy skin lesions. It redefined the segmentation process for categorizing clinical dermoscopic characteristics within superpixels. This model achieved 98.0% accuracy.

A mask RCNN-based model was proposed by Khan et al. [43]. The decorrelation formulation algorithm was used to perform the initial preprocessing of the dermoscopy images. It forwarded the obtained pictures to the MASK-RCNN for lesion segmentation. In this step, the segmented RGB pictures are produced from the ground truth images of the ISIC datasets, and the MASK RCNN model is trained. The DenseNet deep model was given the segmented images as a response to extract features from [44]. It used a Mobile Net model that was transfer-learned and fine-tuned on 10,015 dermoscopy pictures from the HAM10000 dataset after being pre-trained on roughly 1,280,000 images from the 2014 ImageNet Challenge. Figure 8 displays the outcomes of the proposed model and various models based on the same dataset (HAM10000). These comparisons demonstrate that, in performance matrices, the proposed model outperforms CNN, Ensemble Bayesian, and Mobile Net approaches.

While the proposed model exhibits superior performance compared to other studies, it has certain limitations. Specifically, some users may find the YOLOv5 and ResNet models difficult to use due to their demanding computational requirements. Additionally, these models are considered black box models, which means it is impossible to understand how they make their predictions. This can be a limitation for some users who want to understand why the model made a particular prediction.

## 6. Conclusions

The early detection and drastic treatment of melanoma are challenging for professionals, and sometimes, even when presented with identical dermoscopy photos, different experts may reach different results. As a result, the study of skin cancer classification significantly impacts skin cancer secondary diagnoses. To analyze skin lesion image data, this paper primarily examined the categorization of skin lesion images using the HAM10000 database. It contains many challenges, such as the similarity between classes of skin lesions, low contrast, and hair, which appear in some images. The proposed model is based on a small model from the YOLOv5 and ResNet networks. To classify seven skin lesions and detect melanoma, a bounding box provided with probability was used. The model consists of three stages to obtain the best categorization accuracy possible: preprocessing, hyperparameters, additional layers, previewing, and annotating images. The third stage assigns labels with probability classes to each image for the diagnosis. Finally, the average performance metrics are 99.0%, 98.6%, 98.8%, 98.3%, and 98.7% for precision, recall, DSC, MAP from 0.0 to 0.5, and MAP from 0.5 to 0.95, respectively. Along with the recent studies for skin cancer diagnoses, the researchers hope to increase their success in future work by enhancing the model with patients’ individualized data like genes and color. Additional melanoma types and bigger datasets are desperately needed. Additionally, generalizable outcomes are required to test the model against a broader range of skin conditions and make the application practical in most health organizations. It helps doctors, especially undertrained doctors, with guidelines to determine which classes are found and with each probability to determine the degree of disease. It also saves time compared with traditional methods. It assists patients in making self-examinations for guidance and follows their status and treatment.

## Figures and Tables

**Figure 1 diagnostics-13-02804-f001:**
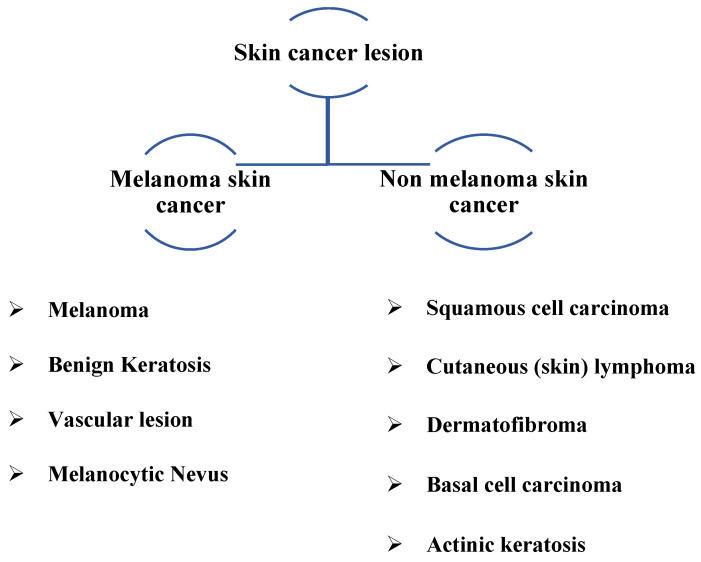
Skin lesion classification with common types.

**Figure 2 diagnostics-13-02804-f002:**
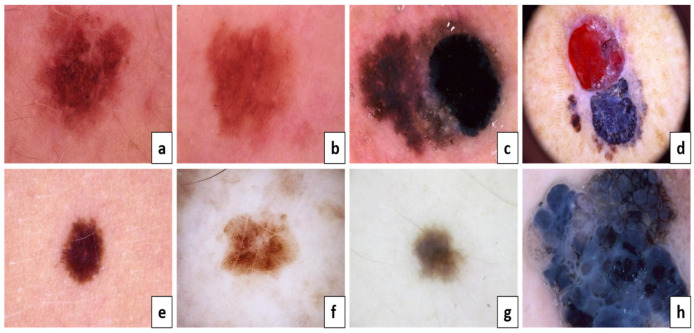
The ABCD lesion diagnosis criteria focus on identifying specifics. Asymmetry: (**a**) both sides match the other, and (**b**) one side does not match the other. Border: (**c**) regular edges and (**d**) irregular or blurred. Color: (**e**) consistent shades and (**f**) different shades. Diameter: (**g**) the lesion is smaller than 6 mm and (**h**) the lesion is larger than 6 mm.

**Figure 3 diagnostics-13-02804-f003:**
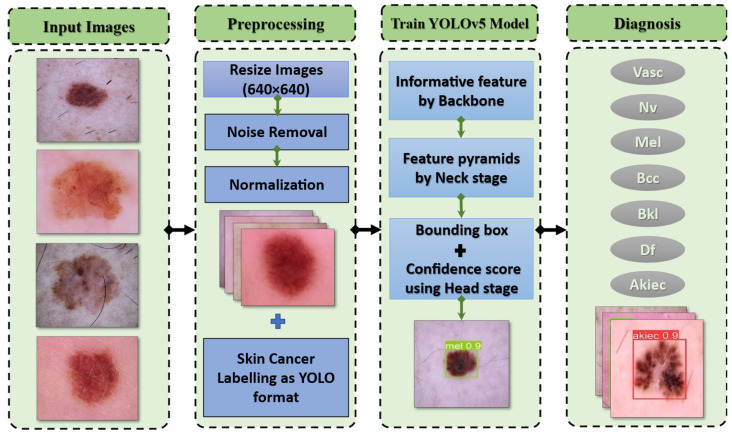
The proposed framework for categorizing seven skin lesions.

**Figure 4 diagnostics-13-02804-f004:**
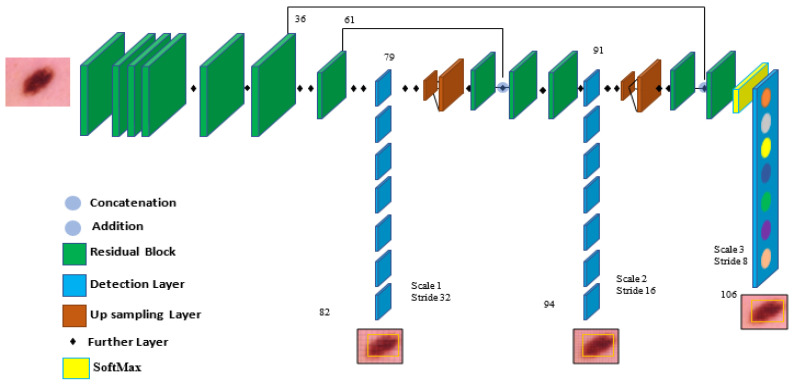
The components of the YOLOv5 model are used for melanoma classification.

**Figure 5 diagnostics-13-02804-f005:**
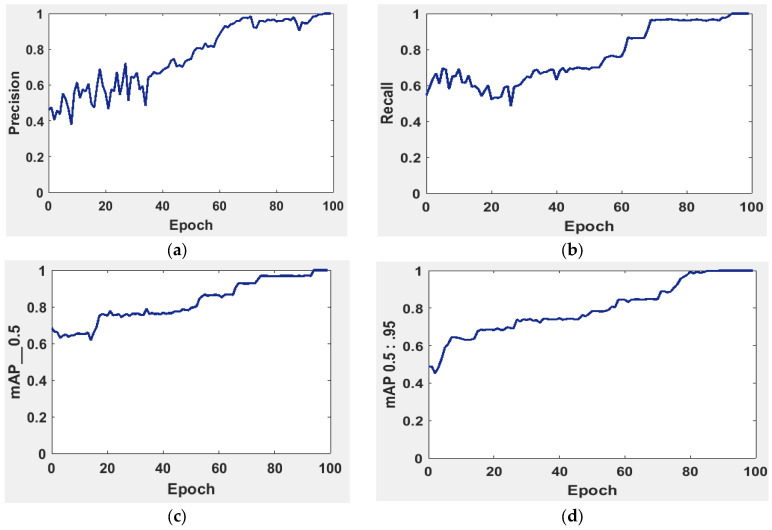
The evaluation of the precision curve, recall curve, and mean average precision at two thresholds for the second run using 100 epochs. (**a**) precision curve, (**b**) recall curve, (**c**) mean average precision at threshold equal 0 to 0.5, and (**d**) mean average precision at threshold equal 0.5 to 0.95.

**Figure 6 diagnostics-13-02804-f006:**
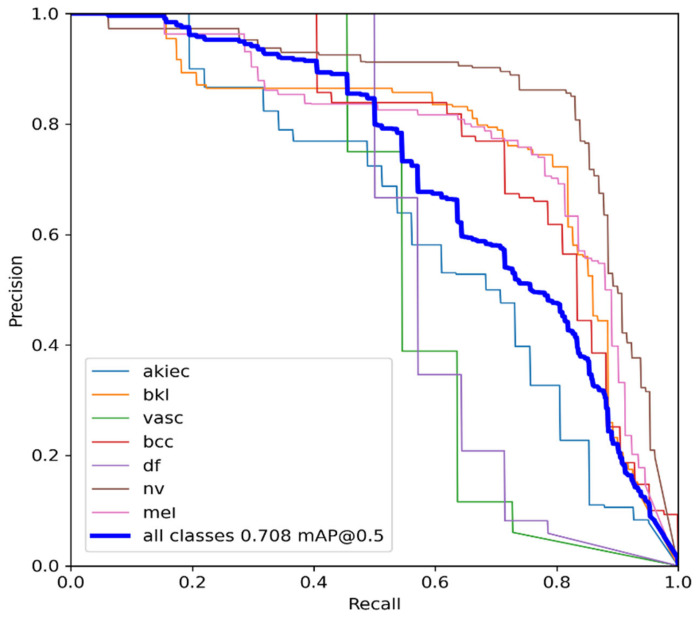
YOLOv5 precision–recall curves for each class. The average precision for each class is the area under each curve for the HAM10000 dataset.

**Figure 7 diagnostics-13-02804-f007:**
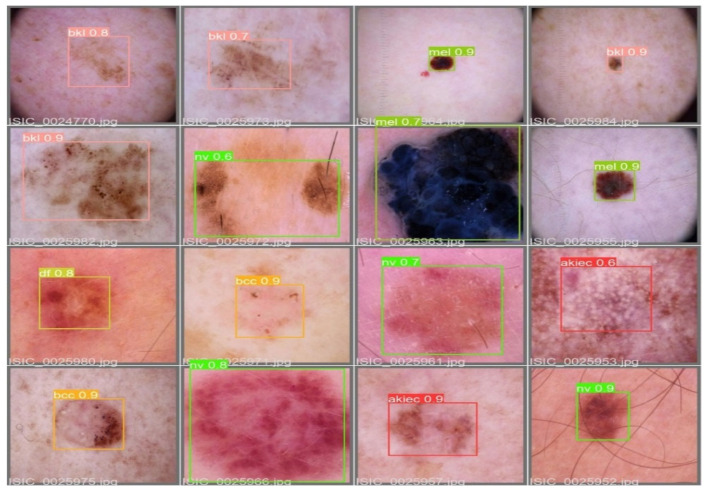
An example batch of detection results on some test images using YOLOv5.

**Figure 8 diagnostics-13-02804-f008:**
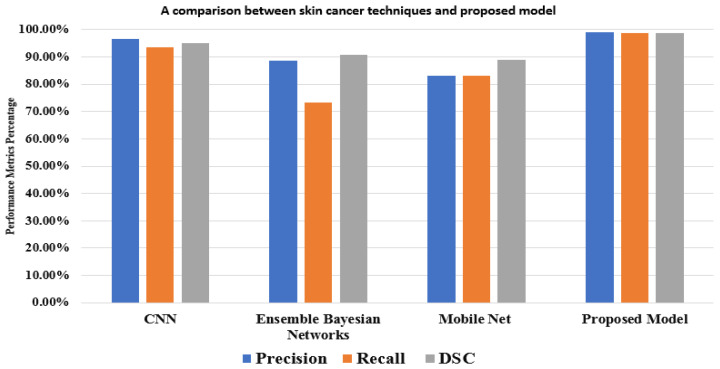
A comparison between the proposed model and other techniques based on the HAM10000 dataset.

**Table 1 diagnostics-13-02804-t001:** Some examples of recent studies on related topics.

Reference	Proposed Technique	Accuracy	Limitation
Premaladha et al. [6]	Segmentation using Otsu’s normalized algorithm and then classification	SVM (90.44), DCNN (92.89), and Hybrid AdaBoost (91.73)	Uses only three classes of skin cancer lesions
Codella et al. [10]	Melanoma recognition using DL, sparse coding, and SVM	93.1%	Need to deepen features and add more cases of melanoma
Waheed et al. [12]	Diagnosing melanoma using the color and texture of different types of lesions	SVM (96.0%)	Need more attributes of skin lesions
Hekler et al. [20]	Classifying histopathologic melanoma using DCNN	68.0%	Uses low resolution and cannot differentiate between melanoma and nevi classes
Pham et al. [21]	Classification using DCNN	AUC (89.2%)	Less sensitivity
Yu et al. [22]	Segmentation and classification using DCNN and FCRN	AUC (80.4%)	Insufficient training data
Li and Shen [23]	Two FCRN for melanoma segmentation and classification	AUC (91.2%)	Overfitting in AUC and low segmentation
Seeja and Suresh [24]	Segmenting data using form, color, and texture variables, then classification using SVM, RF, KNN, and NB	SVM (85.1%), RF (82.2%), KNN (79.2%), and NB (65.9%)	Low classification accuracy
Nasiri et al. [25]	Using the 19-layer model of CNN for melanoma classification	75.0%	Need to enhance accuracy

**Table 2 diagnostics-13-02804-t002:** The distribution of the HAM10000 dataset for training and testing sets.

	VASC	NV	MEL	DF	BKL	BCC	AKIEC
All images	142	6705	1113	115	1099	514	327
Train	115	5360	891	92	879	300	262
Test	27	1345	222	23	220	214	65

**Table 3 diagnostics-13-02804-t003:** Google Colab software requirements and its versions.

Requirement	Version	Requirement	Version
Base	matplotlib ≥ 3.2.2	Export	coremltools ≥ 6.0 by CoreML export
	opencv-python ≥ 4.1.1		onnx ≥ 1.9.0 by ONNX export
	Pillow ≥ 7.1.2		onnx-simplifier ≥ 0.4.1 by ONNX simplifier
	PyYAML ≥ 5.3.1		Nvidia-pyinde by TensorRT export
	requests ≥ 2.23.0		nvidia-tensorrt by TensorRT export
	scipy ≥ 1.4.1		scikit-learn ≤ 1.1.2 by CoreML quantization
	torch ≥ 1.7.0		tensorflow ≥ 2.4
	tqdm ≥ 4.64.0		tensorflowjs ≥ 3.9.0 by TF.js export
	protobuf ≤ 3.20.1		openvino-dev by OpenVINO export
Plotting	pandas ≥ 1.1.4	Extras	ipython by interactive notebook
	seaborn ≥ 0.11.0		psutil by system utilization
Logging	tensorboard ≥ 2.4. clearml ≥ 1.2.0		thop ≥ 0.1.1 by FLOPs computation
			by albumentations ≥ 1.0.3
			by pycocotools ≥ 2.0

**Table 4 diagnostics-13-02804-t004:** Training guidelines for the first and second experiments with 640 image size.

Parameter	First Run	Second Run	Definition
Epoch	300	100	The frequency with which the learning algorithm operates
Batch_size	16	32	How many training instances are used in a single iteration
lr0	0.001	0.001	Initial learning rate (SGD = 1 × 10^−2^; Adam = 1 × 10^−3^)
Lrf	0.2	0.2	Final OneCycleLR learning rate (lr0 × lrf)
Momentum	0.937	0.937	SGD momentum/Adam beta1
warmup_epochs	3.0	3.0	Warmup epochs (fractions ok)
weight_decay	0.0005	0.0005	Optimizer weight decay, 5 × 10^−4^
warmup_momentum	0.8	0.8	Warmup initial momentum
warmup_bias_lr	0.1	0.1	Warmup initial bias learning rate
Box	0.05	0.05	Box loss gain
Cls	0.5	0.5	Class loss gain
cls_pw	1.0	1.0	Cls BCELoss positive_weight
Obj	1.0	1.0	Obj loss gain (scale with pixels)
obj_pw	1.0	1.0	Obj BCELoss positive_weight
anchor_t	4.0	4.0	Anchor-multiple threshold
iou_t	0.20	0.20	IOU training threshold
Scale	0.5	0.5	Image scale (+/−gain)
Shear	0.0	0.0	Image shear (+/−deg)
Perspective	0.0	0.0	Image perspective (+/−fraction), range 0–0.001

**Table 5 diagnostics-13-02804-t005:** Utilizing 300 epochs, the model YOLOv5s results on the HAM10000 dataset.

	Precision (%)	Recall (%)	DSC (%)	MAP 0.0:0.5 (%)	MAP 0.5:0.95 (%)	Accuracy (%)
AKIEC	99.1	94.9	96.9	99.7	95.2	95.2
BKL	95.3	96.8	96.0	95.3	94.5	96.1
VASC	97.0	95.6	96.2	98.7	95.5	97.2
BCC	97.1	97.6	97.3	97.5	96.4	97.3
DF	98.7	99.5	99.0	94.3	94.8	98.8
NV	100.0	98.6	99.2	96.4	99.5	98.1
MEL	98.8	100.0	99.3	98.2	98.6	100.0
Average	98.1	97.5	97.7	97.1	96.3	97.5

**Table 6 diagnostics-13-02804-t006:** Results of the model YOLOv5s for HAM10000 dataset utilizing 100 epochs.

	Precision (%)	Recall (%)	DSC (%)	MAP 0.0:0.5 (%)	MAP 0.5:0.95 (%)	Accuracy (%)
AKIEC	100.0	96.7	98.3	98.9	99.7	98.8
BKL	98.2	98.2	98.2	97.6	94.9	98.9
VASC	98.8	99.6	99.1	97.9	97.9	99.4
BCC	97.1	96.9	96.9	99.5	99.1	99.7
DF	99.6	98.9	99.2	98.6	96.2	100.0
MV	100.0	100.0	100.0	96.2	100.0	99.8
MEL	99.9	100.0	99.9	99.8	98.9	100.0
Average	99.0	98.6	98.8	98.3	98.7	99.5

**Table 7 diagnostics-13-02804-t007:** The comparison of precision, recall, and DSC over some existing models.

Reference	Year	Method	Precision (%)	Recall (%)	DSC (%)	Accuracy (%)	Dataset
Nasiri et al. [25]	2020	KNN	73.0	55.0	79.0	67.0	ISIC dataset
		SVM	58.0	47.0	66.0	62.0	
		CNN	77.0	73.0	78.0	75.0	
Alsaade et al. [38]	2021	CNN	81.2	92.9	87.5	97.5	PH2
Ali et al. [39]	2021	CNN	96.5	93.6	95.0	91.9	HAM10000
Khaledyan et al. [40]	2021	Ensemble Bayesian Networks	88.6	73.4	90.7	83.6	HAM10000
Chang et al. [41]	2022	XGB classifier	97.4	87.8	90.5	94.1	ISIC
Kawahara et al. [42]	2019	FCNN	97.6	81.3	93.0	98.0	ISIC
Khan et al. [43]	2021	Mask RCNN	88.5%	88.5%	88.6%	93.6	ISIC
Chaturvedi et al. [44]	2020	Mobile Net	83.0%	83.0%	89.0%	83.1	HAM10000
Proposed model	2023	YOLOv5 + ResNet	99.0	98.6	98.8	99.5	HAM10000

**Table 8 diagnostics-13-02804-t008:** The details of datasets that are used for performance evaluation.

Database	Description
PH2	It contains the medical assessment, manual segmentation, and identification of several dermoscopic lesions. A collection of 200 dermoscopic photos was completed by dermatologists with expertise. The photos have an 8-bit RGB color depth and a 768 × 560 pixel resolution. It has 40 melanomas, 80 atypical nevi, and 80 normal nevi.
ISIC	A total of 2000 dermoscopic pictures of cancers. It contains malignant and harmless oncological illnesses. A total of 1372 benign nevi, 254 seborrheic keratoses, and 374 melanomas.

## Data Availability

In this study, publicly accessible datasets were examined. https://www.kaggle.com/datasets/kmader/skin-cancer-mnist-ham10000 has these datasets (accessed on 1 May 2023).

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
