# Peer review of "Early Melanoma Detection Based on a Hybrid YOLOv5 and ResNet Technique"

_diagnostics, 2023, doi:10.3390/diagnostics13172804_

Round 1
Reviewer 1 Report
The abstract lacks essential background information on melanoma, which is a specific type of skin cancer, leaving readers unfamiliar with the topic without the necessary context. This background information is crucial to help readers understand the specific focus of the research and its importance in addressing the challenges of early melanoma detection.
The quality of the figures in the manuscript needs improvement to meet the standard of the journal. The current figures appear to be of low resolution
The quality of the paper would significantly improve with the addition of the major contribution of the work in the introduction. Currently, the introduction lacks a clear and concise statement highlighting the unique and novel aspects of the research.
The manuscript claims to use a ResNet+YOLO model for medical modality detection, but it does not clearly demonstrate how its approach is different from existing hybrid ResNet+YOLO models used in previous studies. The authors need to provide a detailed comparison in the introduction or related work section to highlight the unique contributions of their proposed model
Acceptable
Author Response
Responses for Reviewer 1
Manuscript ID: diagnostics-2549855
Type of manuscript: Article
Article Title: Early Melanoma Detection Based on a Hybrid YOLOv5 and ResNet Technique
- We thank the reviewer for his thoughtful review of our work. We have thoroughly re-reviewed the manuscript and corrected any errors we came across.
- Comment:
- The abstract lacks essential background information on melanoma, which is a specific type of skin cancer, leaving readers unfamiliar with the topic without the necessary context. This background information is crucial to help readers understand the specific focus of the research and its importance in addressing the challenges of early melanoma detection.
Response:
- The manuscript was enhanced by updating the abstract section. Specifically, the introduction, problem, aim, methods, and findings were included at the beginning of the abstract, followed by a conclusion at the end.
"Skin cancer, specifically melanoma, is a serious health issue that arises from the melanocytes, the cells that produce melanin, the pigment responsible for skin color. With skin cancer on the rise, timely identification of skin lesions is crucial for effective treatment. However, the similarity between some skin lesions can result in misclassification, which is a significant problem. It's important to note that benign skin lesions are more prevalent than malignant ones, which can lead to overly cautious algorithms and incorrect results. As a solution, researchers are developing computer-assisted diagnostic tools to detect malignant tumors early. First, a new model based on the combination of "you only look once" (YOLOv5) and "ResNet50" is proposed for melanoma detection with its degree using humans against a machine with 10,000 training images (HAM10000). Second, feature maps integrate gradient change, which allows rapid inference, boosts precision, and reduces the number of hyperparameters in the model, making it smaller. Finally, the current YOLOv5 model is changed to get the desired outcomes by adding new classes for dermatoscopic images of typical lesions with pigmented skin. The proposed approach improves melanoma detection with a real-time speed of 0.4 MS non-maximum suppression (NMS) per image. The performance metrics average is 99.0%, 98.6%, 98.8%, 99.5, 98.3%, and 98.7% for precision, recall, dice similarity coefficient (DSC), accuracy, mean average precision (MAP) from 0.0 to 0.5, and MAP from 0.5 to 0.95, respectively. Compared to current melanoma detection approaches, the provided approach is more efficient in using deep features."
- The introduction section has been revised to include statistics highlighting the significance of detecting melanoma early. The updated information is as follows:
â–ª "In 2023, the American Cancer Society predicts that about 97,610 new melanomas will be diagnosed in the country [3]. Melanoma is predicted to be fatal to approximately 7,990 people (about 5,420 men and 2,570 women). Rates of melanoma have considerably increased in recent decades."
- Comment:
- The quality of the figures in the manuscript needs improvement to meet the standard of the journal. The current figures appear to be of low resolution.
Response:
- The figures were enhanced according to your recommended comment.
- Comment:
- The quality of the paper would significantly improve with the addition of the major contribution of the work in the introduction. Currently, the introduction lacks a clear and concise statement highlighting the unique and novel aspects of the research.
Response:
- We have updated the introduction section by including contribution points in the following manner:
"The proposed system has four significant contributions over previous computer-assisted skin cancer screening approaches:
- The proposed method applies to any image (dermoscopy or photographic) of pigmented skin lesions using you only look once (YOLOv5) and ResNet.
- The suggested system classifies samples and determines each class with probability.
- It interacted directly with the skin-color images that were obtained with different sizes.
- The proposed approach enhances scalability since YOLO and ResNet can detect melanoma in huge datasets. This is crucial as it allows for developing more precise and effective melanoma detection systems."
- Comment:
- The manuscript claims to use a ResNet + YOLO model for medical modality detection, but it does not demonstrate how its approach is different from existing hybrid ResNet + YOLO models used in previous studies. The authors need to provide a detailed comparison in the introduction or related work section to highlight the unique contributions of their proposed model.
Response:
- In the related work section, we added a new study based on YOLOv5 at reference [26] and showed its limitations. "This study utilized hyperspectral imaging to examine the ISIC dataset and applied YOLOv5 to identify and categorize various types of skin cancer. The focus of the training phase was on three categories: basal cell carcinoma (BCC), squamous cell carcinoma (SCC), and seborrheic keratosis (SK). However, the model may have developed a bias toward detecting skin cancer within a specific demographic. The accuracy rates for the RGB and HSI models were 79.2% and 78.7%, respectively. The proposed model relied on hyperspectral imaging to eliminate noise, but it was insufficient. Additionally, there are resemblances between BCC and SK, which resulted in confusion between the categories and lower accuracy."

Reviewer 2 Report
The detailed comments on the proposed approach are enlisted in the attached document. Although the authors have written a sound and detailed approach, still, some comments are proposed below to improve the quality of the manuscript.

Author Response
Responses for Reviewer 2
Manuscript ID: diagnostics-2549855
Type of manuscript: Article
Article Title: Early Melanoma Detection Based on a Hybrid YOLOv5 and ResNet Technique
- Thank you so much for your comments. The manuscript was modified to adjust the content.
- In the introduction section, the authors try to elaborate on the motivation for the proposed However, it can further improve by explicitly defining it in the introduction to make it more interesting for the readers.
Response:
- The manuscript was enhanced by including motivations in the introduction section before the challenges, as outlined:
"Melanoma is a serious form of skin cancer that can be deadly. Detecting it in its early stages is crucial for successful treatment and improved health outcomes. The primary goal of melanoma detection is to increase early detection rates, which could potentially save lives. However, it can be difficult for individuals to identify melanoma on their own because it can resemble other skin lesions, such as moles."
- In the related work section, we emphasized the significance of detecting melanoma and highlighted the shortcomings of current studies, as follows:
"Considering the importance of early melanoma identification, the visible similarity among melanoma and non-melanoma tumors, the absence of contrast between lesions and skin, and other considerations. Therefore, accurate automatic diagnosis of skin tumors is crucial to improving the precision and effectiveness of pathologists. "
"According to previous studies, there are limitations in melanoma detection due to the small number of classes used and the need to determine the degree of classes. There is a need to deepen feature extraction and add more cases to diagnose melanoma. The similarities between classes can't be determined. Recently, work has been based on two stages to determine melanoma (segmentation and classification). But there is overfitting in AUC, and the segmentation results are low. A system that relies on DL must be created to obtain reliable melanoma classification. YOLOv5 relies on a single step for identifying and classifying skin lesions to determine the type of melanoma, in contrast to earlier DL studies for skin lesion classification that concentrated on employing specific layers."
- The literature review section should mention the importance of deep learning/ transfer learning in medical health Also, update the related section by including some latest papers in health analytics.
A Hybrid Deep Learning-Based Approach for Brain Tumor Classification. DeepBreastCancerNet: A Novel Deep Learning Model for Breast Cancer Detection Using Ultrasound Images
Response:
- In the related work section, we added the suggested references to show the importance of DL in the medical analysis field:
"Through deep learning models, it's possible to train machines to provide personalized treatment, classify medical images (like X-rays), identify new biomarkers, and predict patient outcomes such as the risk of death or the success rate of surgery. Additionally, this technology has the potential to reduce healthcare costs [19,20]. We have provided examples of pertinent DL research in Table 1."
- Normally, we first elaborate on the proposed methodology abstractly and then explain each step of the methodology in detail. The authors have first elaborated on the preprocessing and then the proposed framework is discussed. The presentation of the paper can be improved further by rearranging the various sections of the For example, section 3.2 can be moved first and then explain the pre-processing but before that dataset and experimental setup should be explained first, etc.
Response:
- The paper was reorganized based on your suggested comment. Thank you for your input.
- The authors should discuss the computational complexity of the proposed model (e.g., training runtime, number of trainable parameters). The numbers should be compared with others in the literature as the authors claim other works "have a lot of deep layers and a lot of parameters".
Response:
- In the methodology section, we added a paragraph to show the available versions with the number of parameters and the needed memory size as indeed:
"The proposed framework utilized a small model with only 7MB of memory. Here is a brief explanation of all the versions of YOLOv5, which is used for model configuration. First, YOLOv5n (the nano edition) is the smallest in the series, intended for Internet-of-Things data, and it also supports OpenCV Dl. In INT8 format, it weighs less than 2.5 MB; in FP32 format, it is about 4 MB. It's perfect for software devices. Second, YOLOv5s (the small edition) is the family's smallest model, with about 7.2 million parameters, making it perfect for inference to run on the CPU. Third, this medium-sized version is interesting, unlike YOLOv5m (the medium edition), which has 21.2 million parameters. Given that it offers a pretty good balance between speed and efficiency, it may be the model that is most suitable for many datasets and training. Fourth, YOLOv5l (the large edition) contains 46.5 million components, the biggest version in the YOLOv5 group. For datasets that necessitate us to find tiny things, it works perfectly. Finally, YOLOv5x (the extra-large edition) is the largest of the five and has the greatest MAP, although it has 86.7 million parameters and therefore is slower than the others. But, ResNet has 25.6 million parameters."
- A few more figures demonstrating the efficiency of the proposed algorithm may be The authors may use tSNE or Deep dream images for such demonstration. Layer-wise image transformations may also be included.
Response
- We have already utilized this network using the identical version (YOLOv5 small size).
Identification and Classification of Crowd Activities
- Although a discussion section is added in the paper, it is focusing on the comparison of the proposed approach with state-of-the-art. Usually, in the discussion section, we emphasize highlighting the advantages of the proposed study, practical implications, and possible social implications of the study. The discussion section should be rewritten to improve the paper's
Response:
- The discussion section was rewritten according to your comment as follows:
"The early detection of melanoma is crucial for improving treatment and prognosis. Screening a vast number of images for melanoma can be achieved using YOLOv5 and ResNet, which can enhance early detection rates in real time. Through this, more lives can be saved, and the quality of life for melanoma patients can be improved. A comparison analysis is presented in this section to assess the performance of the suggested model as a melanoma detection approach utilizing modified YOLOv5 and ResNet techniques. In Table. 7, our model is compared with other models. Despite the similarities between classes, the suggested YOLOv5 has detected melanoma using adequate coordinates, including the bounding box. The comparison results allow us to conclude that the suggested YOLOv5 model is reliable for melanoma detection in real-time photos that have been gathered."
- The proposed approach lacks limitations of the proposed study. Clarifying the study's limitations allows the readers to better understand under which conditions the results should be A clear description of the limitations of a study also shows that the
researcher has a holistic understanding of his/her study. However, the authors fail to demonstrate this in their paper.
Response:
- The paper's discussion section presented limitations, followed by a conclusion section containing future work and practical advantages of automated melanoma detection for health organizations.
"While the proposed model exhibits superior performance compared to other studies, it has certain limitations. Specifically, some users may find the YOLOv5 and ResNet models difficult to use due to their demanding computational requirements. Additionally, these models are considered black box models, which means it is impossible to understand how they make their predictions. This can be a limitation for some users who want to understand why the model made a particular prediction."
- Time complexity should be
Response:
- As we mentioned in the methodology section:
"The proposed framework utilized a small model with only 7MB of memory. Here is a brief explanation of all the versions of YOLOv5, which is used for model configuration. First, YOLOv5n (the nano edition) is the smallest in the series, intended for Internet-of-Things data, and it also supports OpenCV Dl. In INT8 format, it weighs less than 2.5 MB; in FP32 format, it is about 4 MB. It's perfect for software devices. Second, YOLOv5s (the small edition) is the family's smallest model, with about 7.2 million parameters, making it perfect for inference to run on the CPU. Third, this medium-sized version is interesting, unlike YOLOv5m (the medium edition), which has 21.2 million parameters. Given that it offers a pretty good balance between speed and efficiency, it may be the model that is most suitable for many datasets and training. Fourth, YOLOv5l (the large edition) contains 46.5 million components, the biggest version in the YOLOv5 group. For datasets that necessitate us to find tiny things, it works perfectly. Finally, YOLOv5x (the extra-large edition) is the largest of the five and has the greatest MAP, although it has 86.7 million parameters and therefore is slower than the others. But ResNet has 25.6 million parameters. The network has a performance capability of 17 GFlops. Gigaflops is a unit of measurement used to measure the computational power of a computer's floating-point unit."
- Furthermore, the paper lacks in discussing future
Response:
- In the conclusion section of the paper, future work was presented, followed by practical advantages of using automated melanoma detection for health organizations.
"Along with the recent studies for skin cancer diagnosis, the researchers hope to increase their success in future work by enhancing the model with patients' individualized data like genes and color. Additional melanoma types and a bigger dataset are desperately needed. Additionally, generalizable outcomes are required to test the model against a broader range of skin conditions and make the application practical in most health organizations. It helps doctors, especially undertrained doctors, with guidelines to determine which classes are found and each probability to determine the degree of disease. It also saves time compared with traditional methods. It assists patients in self-examinations for guidance and follows their status and treatment."

Round 2
Reviewer 1 Report
Accepted as it is
Reviewer 2 Report
The authors improved the quality of the paper by incorporating all the reviews and suggestions. Therefore, the paper is accepted in its current form.